

# Using a mobile health app to improve patients' adherence to hypertension treatment: a non-randomized clinical trial

Simiane Salete Volpi[1], Daiana Biduski[2], Ericles Andrei Bellei[2], Danieli Tefili[1], Lynn McCleary[3], Ana Luisa Sant'Anna Alves[1] and Ana Carolina Bertoletti De Marchi[1,2]

[1] School of Physical Education and Physiotherapy, University of Passo Fundo, Passo Fundo, RS, Brazil
[2] Institute of Exact Sciences and Geosciences, University of Passo Fundo, Passo Fundo, RS, Brazil
[3] Faculty of Applied Health Sciences, Brock University, St. Catharines, ON, Canada

## ABSTRACT

Poor adherence to hypertension treatment increases complications of the disease and is characterized by a lack of awareness and acceptance of ongoing treatment. Mobile health (mHealth) apps can optimize processes and facilitate access to health information by combining treatment methods with attractive solutions. In this study, we aimed at verifying the influence of using an mHealth app on patients' adherence to hypertension treatment, also examining how user experience toward the app influenced the outcomes. A total of 49 participants completed the study, men and women, diagnosed with hypertension and ongoing medical treatment. For 12 weeks, the control group continued with conventional monitoring, while the experimental group used an mHealth app. From the experimental group, at baseline, 8% were non-adherent, 64% were partial adherents and 28% were adherent to the treatment. Baseline in the control group indicated 4.2% non-adherents, 58.3% partial adherents, and 37.5% adherents. After follow-up, the experimental group had an increase to 92% adherent, 8% partially adherent, and 0% non-adherent ($P < 0.001$). In the control group, adherence after follow-up remained virtually the same ($P \geq 0.999$). Results of user experience were substantially positive and indicate that the participants in the experimental group had a satisfactory perception of the app. In conclusion, this study suggests that using an mHealth app can empower patients to manage their own health and increase adherence to hypertension treatment, especially when the app provides a positive user experience.

# INTRODUCTION

High blood pressure (>140/90 mmHg), also known as hypertension (*Egan & Zhao, 2012*), is a concerning public health issue—one-third of patients with hypertension have not been diagnosed and among those who are diagnosed, about half do not follow treatment properly (*Kitt et al., 2019*). When patients follow the recommendations provided by health professionals, both pharmacological and non-pharmacological, there is an improvement in treatment adherence (*Chudiak, Jankowska-Polańska & Uchmanowicz,*

Corresponding author
Ericles Andrei Bellei, 168729@upf.br

2017; *Uchmanowicz et al., 2018*; *Da Silva et al., 2017*). Adherence is the extent to which a person's behavior-taking medication, following a diet, and/or executing lifestyle changes corresponds with agreed recommendations from a health-care provider (*Sabaté, 2003*). Poor adherence to hypertension treatment is a complex issue affected by multiple factors, including patients' age, educational level, number of prescribed medications, social-economic status, number of comorbidities, lack of awareness and others (*Mancia et al., 2013*; *Miller, 2016*; *Uchmanowicz et al., 2018*; *Ni et al., 2019*). Therefore, the treatment requires permanent motivation to maintain daily health care (*Chudiak, Jankowska-Polańska & Uchmanowicz, 2017*).

Non-pharmacological approaches for hypertension include diet modifications, self-monitoring of blood pressure, and other health behaviors and habits (*Gewehr et al., 2018*), which could all be strengthened with the support of technology. Mobile health apps (mHealth) offer a way to monitor patient's health conditions, such as diet, body weight, blood pressure, mood, and sleep, among others, and can be used in combination with traditional health care to facilitate access to health information (*Hui et al., 2019*; *Ni et al., 2018*; *Alessa et al., 2019*; *Galligioni et al., 2015*; *Albrecht et al., 2017*). Thus, mHealth apps might increase awareness of needed behavioral changes and the adherence to healthy habits, along with the health care provider's awareness of what the patient is doing (*Paglialonga, Lugo & Santoro, 2018*; *Bellei et al., 2020*). Moreover, mHealth apps can guide illness self-management, providing patients with psychological support and decision-making support, and facilitating collaboration between health professionals, patients, and their families (*Lu et al., 2019*).

To improve adherence, patients need frequent encouragement, guidance and reminders about lifestyle-related to hypertension management, monitoring blood pressure symptoms, and health status indicators (*Whelton, 2015*; *Da Silva et al., 2017*). Mobile health apps can remind patients of healthy habits, such as checking blood pressures regularly and taking medications as prescribed, all leading to better treatment adherence (*Kitt et al., 2019*; *Xiong et al., 2018*). However, the use of mHealth interventions also requires long-term studies to understand the real impact of technology in health, and to investigate human factors associated with perceptions and usage (*Dick et al., 2016*; *McLean et al., 2016*, *Toro-Ramos et al., 2017*; *Biduski et al., 2020*).

Regarding human factors, the concept of user experience includes all aspects of interaction and involves interpreting the user's needs, intentions and perspectives, evaluating emotional responses, impressions, and ideas about a product (*Zarour & Alharbi, 2017*). User experience research can be used to clarify the specific needs and goals of mHealth users, thereby providing patients with an opportune healthcare experience (*Kirkscey, 2020*; *Biduski et al., 2020*). Notwithstanding, evaluating the real effects of user experience requires long periods, over extended use (*Karapanos et al., 2010*; *Kajiwara & Jin, 2012*; *Minge & Thüring, 2018*). Long-term approaches are also imperative to evaluate treatment adherence, as this can fall off over time. From this perspective, the primary objective of this study was to test the effect of using an mHealth app on patients' adherence to hypertension treatment. The secondary objective was to examine how patients' user experience might have influenced the outcomes.

## METHODS

We conducted a quasi-experimental study (non-randomized, controlled, open-label) and collected participants' data at enrollment and 12 weeks after an mHealth intervention. This time frame was based on the study of *Neumann et al. (2015)*, which justified that at least 3 months were needed to notice long-term effects. This was also considered the minimum period to assess the effects on the user experience (*Karapanos et al., 2010*; *Kajiwara & Jin, 2012*; *Minge & Thüring, 2018*). The local ethics committee of the University of Passo Fundo, under opinion number 3,414,793, approved all procedures involving humans. Written informed consent was obtained from all participants. Registration occurred on the Brazilian Registry of Clinical Trials, code RBR-2rkkgn. This study is a secondary part of a larger multidisciplinary project for health innovation funded by the National Council for Scientific and Technological Development—CNPq and the Ministry of Health of Brazil. The project aims to develop and test a comprehensive electronic health platform to be made available to the Brazilian public health system (*De Marchi et al., 2020*).

### Sample and allocation

A total of 74 volunteer participants (of whom 49 completed the study) were recruited by phone calls. They were men and women aged 24 to 69 years, diagnosed with arterial hypertension, who were receiving ongoing medical treatment at primary health centers in the city of Passo Fundo, Rio Grande do Sul, Brazil. We attended two of these centers to recruit participants on a convenience basis.

The sample size for this study was based on the sample size that will be recruited in the definitive clinical trial of the main project, whose reasoning about estimated effects is detailed in the protocol by *De Marchi et al. (2020)*. Eligibility criteria were: (1) current and ongoing medical monitoring regarding hypertension treatment; (2) minimum score on MMSE cognitive screening test (*Brucki et al., 2003*); (3) ability to have measurement of blood pressure periodically from an electronic blood pressure cuff or a sphygmomanometer. In addition, participants allocated to the experimental group were required to have (1) familiarity with the use of smartphone apps; (2) a smartphone with Android operating system version 5 or higher; (3) Internet access on the smartphone.

The allocation of participants was determined by meeting the final three eligibility criteria. For instance, if the participant met all the criteria but did not have a compatible mobile phone to use the app, then the participant would be allocated to the control group. If the participant had a compatible mobile phone, they would be allocated to the experimental group.

### Measurements

At baseline, we collected basic demographic data from all participants: gender, marital status, age, education level, and monthly household income. Adherence to hypertension treatment was measured at baseline and after follow-up using the Martín-Bayarre-Grau (MBG) questionnaire (*Matta, Luiza & Azeredo, 2013*; *Alfonso, Vea & Ábalo, 2008*).

This validated instrument is a cross-cultural adaptation from its original version, which determines the level of adherence according to the operational definition of therapeutic adherence formulated by WHO. The questionnaire includes information about the patient's medication, doctor appointments, treatment, diet, and exercise. It consists of 12 statements answered on a five-point Likert scale (never, almost never, sometimes, almost always and always). The higher score means greater adherence. Participants were classified as "adherent" if they obtained 38 to 48 points, "partial adherent" if they obtained 18 to 37 points and "non-adherent" from 0 to 17 points (*Alfonso, Vea & Ábalo, 2008*).

User experience in the experimental group was evaluated using the User Experience Questionnaire (UEQ) (*Laugwitz, Held & Schrepp, 2008*) after follow-up. UEQ is a validated instrument composed of 26 items with a semantic differential rating scale of seven points. The items are related to the six user experience scales of attractiveness, perspicuity, efficiency, dependability, stimulation, and novelty. Attractiveness is a pure valence dimension. Perspicuity, efficiency, and dependability are pragmatic quality aspects (goal-directed), while stimulation and novelty are hedonic quality aspects (not goal-directed) (*Zarour & Alharbi, 2017*). From responses in seven semantic differential points, the scores generated by the UEQ range from −3.0 to 3.0. A result less than −0.8 indicates a negative user experience; a result between −0.8 and 0.8 indicates a neutral user experience; a result greater than 0.8 indicates a positive user experience.

## Procedure and follow-up

We instructed participants from both groups to continue their hypertension treatment as usual. For the experimental group, we first installed the mHealth app on the participants' smartphones. Then, we created a user account for each participant and instructed them on how to use the app to record data such as blood pressure and other measurements. During the recruitment process, we helped participants who had difficulty using the app and gave them feedback and more detailed instructions. Health professionals were also made available to assist participants remotely in using the app. Participants could request help from professionals through a text-based chat functionality available on the app.

In the follow-up, the control group had 32 participants who continued their conventional hypertension treatment, without any contact with the app during the study. The experimental group had 36 participants who completed the study using the app for 12 weeks. The mHealth app was developed by *Cechetti et al. (2019)*. It has elements designed to engage patients in self-monitoring of health conditions. It includes the recording of variables (or factors) related to hypertension management, including blood pressure, weight, waist circumference, height, sleep, mood, and engagement in physical activities. Other features include risk assessment based on reference values, recommendations, alerts, and reminders about medication, logbooks of physical activities, and blood pressure measurements. All these elements incorporated in the app are related to a healthy lifestyle that facilitates the treatment of hypertension. Patient data is stored in the cloud for integration with a web dashboard, which allows authorized healthcare professionals to remotely monitor the patient.

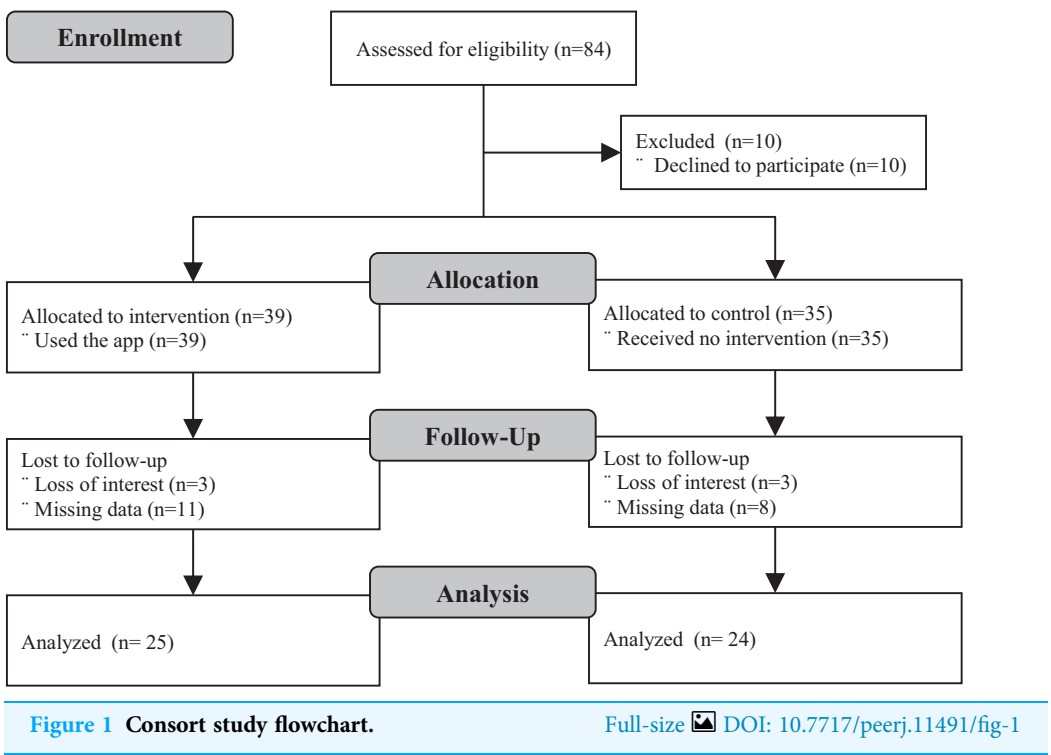

**Figure 1 Consort study flowchart.**

## Statistical analysis

Quantitative data were analyzed using the statistical package SPSS 22.0 (IBM Corporation, Armonk, NY, USA). Descriptive analysis of nominal and ordinal level variables was performed using absolute and relative frequency counts. For continuous variables, we calculated measures of central tendency and dispersion (minimum, maximum, mean, and standard deviation). McNemar's Test was used to test within group differences in treatment adherence. Chi-square test was used to test between group differences. A significance level of 5% was considered for all analyzes. For the MBG questionnaire, we applied the Mann-Whitney test to compare the score between the groups before and after follow-up. We analyzed the UEQ's responses using its proprietary data analysis tool to obtain scores for hedonic and pragmatic qualities and six user experience scales.

## RESULTS

Of the 74 enrolled participants, 49 completed follow-up, 24 in the control group and, 25 in the experimental group. Loss to follow-up was due to participants' loss of interest and inability to provide accurate data for analysis. Enrollment and follow-up took place between August and November 2019. Figure 1 shows the study Consort flowchart and Table 1, the demographic characteristics of participants. The mean and standard deviation of participants' age in the control group was 60.4 ± 10.4 years and 57.2 ± 7.1 years in the experimental group. No statistically significant differences were observed between the experimental and control groups across any of the baseline demographic variables.

Figure 2 presents the overall results of the assessment of adherence to hypertension treatment before (baseline) and after follow-up. From the baseline to the after follow-up,

| Table 1 Baseline demographic characteristics of participants ($N = 49$). | | | |
|---|---|---|---|
| Characteristic | Experimental ($n = 25$) | Control ($n = 24$) | P |
| **Gender, n (%)** | | | 0.056 |
| Male | 15 (60.0) | 8 (33.3) | |
| Female | 10 (40.0) | 16 (66.7) | |
| **Age, n (%)** | | | 0.080 |
| to 29 years | 0 (0.0) | 1 (4.2) | |
| to 49 years | 4 (16.0) | 2 (8.3) | |
| to 59 years | 10 (40.0) | 3 (12.5) | |
| to 69 years | 11 (44.0) | 16 (66.7) | |
| to 79 years | 0 (0.0) | 2 (8.3) | |
| **Marital status, n (%)** | | | 0.475 |
| Single | 4 (16.0) | 3 (12.5) | |
| Married/stable relationship | 16 (64.0) | 14 (58.3) | |
| Divorced | 2 (8.0) | 3 (12.5) | |
| Widow(er) | 3 (12.0) | 4 (16.7) | |
| **Years of study, n (%)** | | | 0.238 |
| to 4 | 1 (4.0) | 4 (16.7) | |
| to 8 | 9 (36.0) | 11 (45.8) | |
| to 11 | 11 (44.0) | 5 (20.8) | |
| or more | 4 (16.0) | 4 (16.7) | |
| **Is retired, n (%)** | | | 0.232 |
| Yes | 12 (48.0) | 15 (62.5) | |
| No | 13 (52.0) | 9 (37.5) | |
| **Household income, n (%)** | | | 0.103 |
| None | 0 (0.0) | 4 (16.7) | |
| Up to 1 minimum wage | 7 (28.0) | 5 (20.8) | |
| to 3 minimum wages | 17 (68.0) | 12 (50.0) | |
| to 5 minimum wages | 1 (4.0) | 3 (12.5) | |

the number of adherents in the experimental group increased substantially. Meanwhile, the numbers after follow-up in the control group remained essentially the same. Table 2 presents the tests of differences in adherence to hypertension treatment inter-group and intra-group. Before follow-up, there was no significant difference between groups. After follow-up, the experimental group had a higher prevalence of treatment adherence than the control group. Within groups, treatment adherence was significantly higher at 12 weeks than at baseline for the experimental group but not in the control group.

Table 3 presents the results of the User Experience Questionnaire. The distribution of responses for each item in the questionnaire is illustrated in Fig. 3. All results were substantially positive and indicated that the participants in the experimental group had a very positive perception toward the app. Considering these findings, we assumed this satisfactory experience influenced the improvements in adherence of participants from the experimental group.
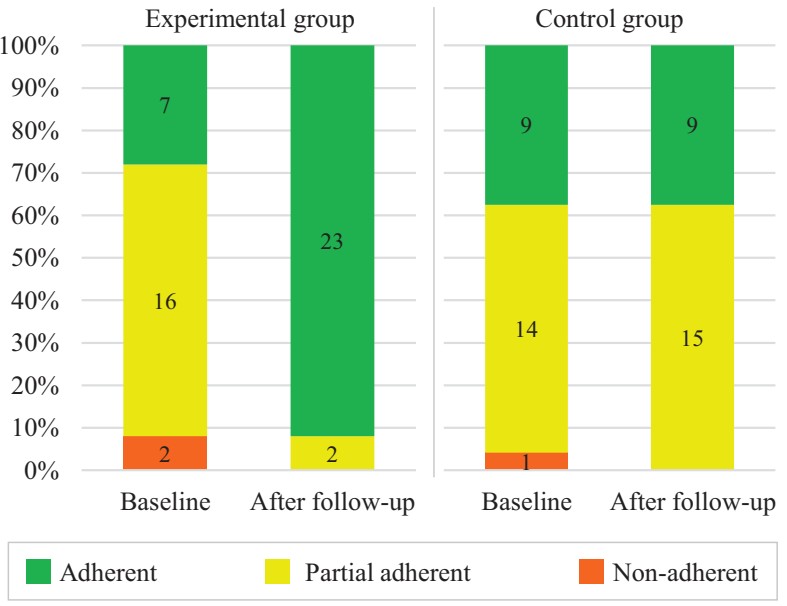

**Figure 2** Assessment of adherence to hypertension treatment according to the Martin-Bayarre-Grau questionnaire (N = 49).

**Table 2 Comparison of the change in adherence to hypertension treatment inter-groups and intra-groups.**

| Group | Baseline (P = 0.343*) | | After follow-up (P < 0.001*) | | P |
|---|---|---|---|---|---|
| | Non-adherent or partial | Adherent | Non-adherent or partial | Adherent | |
| Experimental, n (%) | 18 (72.0) | 7 (28.0) | 2 (8.0) | 23 (82.0) | <0.001[†] |
| Control, n (%) | 15 (62.5) | 9 (37.5) | 15 (62.5) | 9 (37.5) | ≥0.999[†] |

Notes:
  [*] Comparison of changes in adherence to hypertension treatment inter-groups using Chi-square Test.
  [†] Comparison of changes in adherence to hypertension treatment intra-groups (baseline versus after follow-up) using McNemar's Test of paired samples.

# DISCUSSION

The control of hypertension is often complex, as it encompasses a variety of factors, ranging from individual aspects of access, difficulties in seeking health services, acquiring medications, following medical prescription, and adaptations to improve lifestyle changes (*Macinko, Leventhal & Lima-Costa, 2018*; *Ferretto et al., 2020*). As in this study, others show that supporting mHealth apps has the potential to increase adherence to treatment of patients with hypertension (*Santo & Redfern, 2019*; *Parati et al., 2017*).

Health apps can help patients self-manage their health conditions, improve self-assessment, treatment, and control of high blood pressure, including features to collect treatment monitoring data (*Lu et al., 2019*; *Liang et al., 2018*). Several studies state that participants who used technological resources as an intervention obtained better results when compared to the control group (*Debon et al., 2020*; *Morawski et al., 2018*; *Andre, Wibawanti & Siswanto, 2019*; *Márquez Contreras et al., 2019*), similar to our results.

**Table 3 User experience of participants from experimental group ($n$ = 25).**

| Category | Mean* | SD | Variance | 95% CI |
|---|---|---|---|---|
| Pragmatic aspects | 2.60 | 0.06 | 0.42 | 0.20 |
| Hedonic aspects | 2.32 | 0.47 | 0.43 | 0.21 |
| Attractiveness | 2.72 | 0.38 | 0.41 | 0.15 |
| Perspicuity | 2.53 | 0.59 | 0.33 | 0.23 |
| Efficiency | 2.64 | 0.50 | 0.49 | 0.20 |
| Dependability | 2.63 | 0.47 | 0.43 | 0.18 |
| Stimulation | 2.65 | 0.47 | 0.25 | 0.19 |
| Novelty | 1.99 | 0.64 | 0.60 | 0.25 |

Note:
* Results range from −3 to 3. A value greater than 0.8 indicates a positive and satisfactory user experience.

Furthermore, apps facilitate communication between patients and healthcare professionals and contribute to patient education (*Santo & Redfern, 2019*; *Debon et al., 2019*).

In our study, participants who used the app more actively were between 50 and 69 years old. Hypertension is one of the most prevalent chronic conditions associated with age (*Desjardins-Crépeau & Bherer, 2016*). Older adults face many health challenges as they age and generally require a relatively large volume of health services (*Institute of Medicine of The National Academies, 2008*). Conversely, *Daniel & Veiga (2013)* observed that increasing age was associated with a higher probability of adherence to the recommended treatment. Likewise, *Jardim et al. (2017)* found better awareness and control in elderly patients. In this sense, the age of participants in this study may have positively influenced adherence.

*Akoko et al. (2017)* found that factors related to the patient and the health service providers (e.g., regular clinic attendance and condition explanation) showed associations with adherence. Similarly, *Duan et al. (2020)* affirm the support and guidance of health professionals are reasons for high adherence rates. In our study, we assume that the high adherence to the treatment in the experimental group was also due to the health professionals, who were remotely available to give feedback, assist patients on how to use the app's features, and clarify doubts and concerns that users had about their treatment when necessary. Consequently, these users were more adherent to the treatment, since they received the desired care and felt they were well informed about their condition, as mentioned by *Jankowska-Polańska et al. (2016)*. This type of resource is also fundamental to pave the adoption of telemedicine and remote monitoring technologies, which are playing an increasingly important role for health services amidst the recent pandemic crisis (*Wosik et al., 2020*; *Smith et al., 2020*). Further research is needed to investigate how health professionals and patients can integrate technological advances into daily practice, ensuring the most beneficial and appropriate aspects of technology are effectively used in the health system (*Rowland et al., 2020*).

Initially, users expressed concern about access, lack of trust, and reduced ability to deal with technology (*Albrecht et al., 2017*). However, over time, most users realized that through the support provided by the app they could solve problems, reduce insecurities,
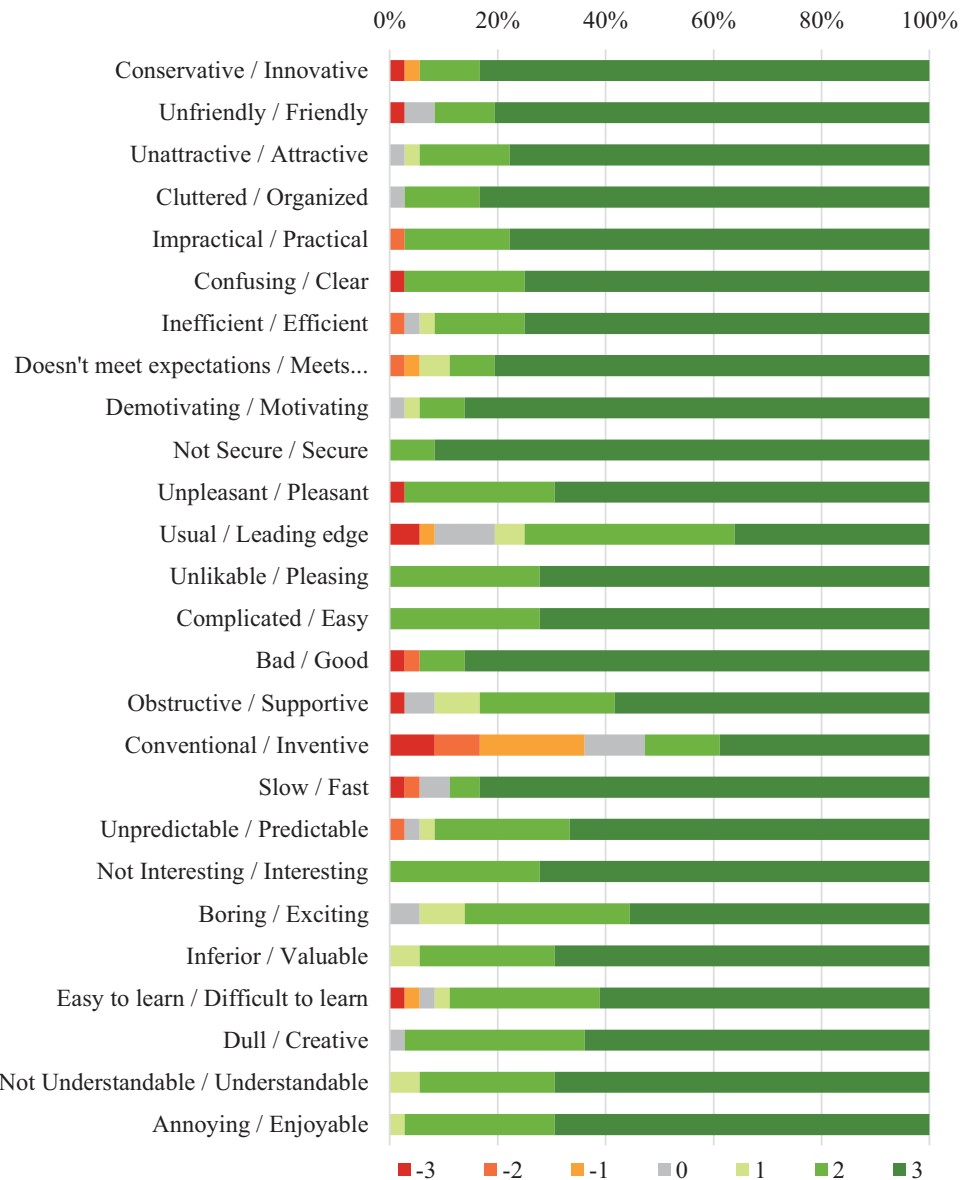

**Figure 3 Distribution of answers per item to the User Experience Questionnaire assessing the app usage in the experimental group.**

and improve their self-monitoring, thus making the time spent with the app pleasant and enjoyable, partly due to the intrinsic rewards provided by the gamification system included in the app (*Cechetti et al., 2019*). These results show that game elements complement the user experience, improving engagement, motivating patients, caregivers, and family members in the quest to acquire more knowledge and use technology to improve health condition (*Da Silva Júnior et al., 2021*). Hence, as long as the quality of health services can meet the expectations of patients, they will continue to use the services (*Guo et al., 2020*).

The results of the UEQ questionnaire summarized participants' impressions regarding the influence of experience on the use of the health app. The UEQ categories that obtained the highest means were attractiveness, efficiency, stimulation, and the category related to pragmatic aspects. Two characteristics include the aspects that can influence the user experience, the pragmatic quality, related to the execution of a certain task; and hedonic quality, related to the intrinsic values of each user and their perceptions (*Zarour & Alharbi, 2017*). In this study, the category related to the pragmatic aspects obtained a higher mean value than the hedonic aspects. We assume the users were more focused on the perceived usefulness of the app than on sentimental value because they believed it was a useful and practical tool for their needs (*Roman et al., 2020*). Likewise, *Dou et al. (2017)* affirm the connection with health professionals and the perceived health threat had significant effects on the perceived usefulness of the participants. According to *Anderson, Burford & Emmerton (2016)*, perceived benefits can improve users' engagement with a health app, and investigating the variety of user experiences and expectations can contribute to the research and design of healthcare apps by encouraging persistence in self-monitoring. If users report a satisfactory experience and rate the functionality and content as useful, then they will be inclined to use the application for a longer period (*Biduski et al., 2020*). Therefore, this study stress how important it is to consider interaction aspects and design implications to provide a better user experience in mHealth apps. Future studies can better explore and statistically test the association between the design aspects of mHealth with the health outcomes resulting from its usage.

This study is a part of a comprehensive multidisciplinary project, in which several complementary studies have already been developed, with the participation of researchers, professors, students, and professionals from different research areas. The main project aims to develop the platform for monitoring hypertension and implement it in the public health network in Brazil (*De Marchi et al., 2020*). The multiple studies conducted to address different perspectives and health outcomes, so that, based on the gathered evidence and other findings, we can provide a valid and effective solution that is capable of improving the quality of health services for patients and health professionals.

## Limitations

This study has some limitations. The assessment of adherence using the Martín-Bayarre-Grau questionnaire relies on participants' self-report and reminiscence, which are naturally susceptible to omissions and misinterpretations during the interview. Regarding the quasi-experimental design, non-randomization may have had a bias influence on the findings, as well as the impossibility of blinding health professionals and interviewers. Nevertheless, all those involved in the study underwent training and used procedure protocols to lessen the possibility of bias. Due to the characteristics of the data for user experience assessment, we did not have a benchmark between groups and were unable to statistically verify the association between user experience scores and the categorical variables of treatment adherence.

## CONCLUSION

Using an mHealth app can empower patients to manage their health and increase adherence to hypertension treatment, especially when the app provides a positive user experience. As better adherence implies several underlying improvements related to the diverse treatment factors, this can be considered a cornerstone of the success of digital interventions. Patients who are satisfied with the app's features on managing their health condition will feel more involved in this process. Thus, the user experience when interacting with the app will be more satisfactory and the patient will be more likely to engage in self-monitoring and comply with the treatment recommendations.

### Funding

This study was supported by the National Council for Scientific and Technological Development—CNPq, Ministry of Health of Brazil—MoH, and Ministry of Science, Technology and Innovation of Brazil—MCTI (conjoint grant number 440078/2018-0). The funders had no role in study design, data collection and analysis, decision to publish, or preparation of the manuscript.

### Grant Disclosures

The following grant information was disclosed by the authors:
National Council for Scientific and Technological Development—CNPq.
Ministry of Health of Brazil—MoH.
Ministry of Science, Technology and Innovation of Brazil—MCTI: 440078/2018-0.

### Competing Interests

The authors declare that they have no competing interests.

### Author Contributions

- Simiane Salete Volpi conceived and designed the experiments, performed the experiments, analyzed the data, prepared figures and/or tables, authored or reviewed drafts of the paper, and approved the final draft.
- Daiana Biduski performed the experiments, analyzed the data, authored or reviewed drafts of the paper, and approved the final draft.
- Ericles Andrei Bellei analyzed the data, prepared figures and/or tables, authored or reviewed drafts of the paper, and approved the final draft.
- Danieli Tefili performed the experiments, analyzed the data, authored or reviewed drafts of the paper, and approved the final draft.
- Lynn McCleary analyzed the data, authored or reviewed drafts of the paper, and approved the final draft.
- Ana Luisa Sant'Anna Alves conceived and designed the experiments, analyzed the data, prepared figures and/or tables, authored or reviewed drafts of the paper, and approved the final draft.

- Ana Carolina Bertoletti De Marchi conceived and designed the experiments, analyzed the data, authored or reviewed drafts of the paper, and approved the final draft.

## Human Ethics
The following information was supplied relating to ethical approvals (i.e., approving body and any reference numbers):

The local ethics committee of the University of Passo Fundo, under opinion number 3.414.793, approved all procedures involving humans.

## Clinical Trial Ethics
The following information was supplied relating to ethical approvals (i.e., approving body and any reference numbers):

The local ethics committee of the University of Passo Fundo, under opinion number 3.414.793, approved all procedures involving humans.

## Data Availability
The raw dataset is available in the Supplemental File.

## Clinical Trial Registration
The following information was supplied regarding Clinical Trial registration:

Brazilian Registry of Clinical Trials RBR-2rkkgn.

## Supplemental Information
Supplemental information for this article can be found online at http://dx.doi.org/10.7717/peerj.11491#supplemental-information.

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
