# Peer review of "Using a mobile health app to improve patients’ adherence to hypertension treatment: a non-randomized clinical trial"

_PeerJ, doi:10.7717/peerj.11491_

## Round 0.1 · original submission · Major Revisions

Your work has been reviewed by three experts in the topic and all of them have indicated scientific merit in their reports. However, there are some major issues which you should address in a revised version of the text. Please, see the comments below so as to have more information.

Reviewer 1 ·

Basic reporting

No comments

Experimental design

This study used quasi experimental design. It is well conducted, but there are some issues that I think should be addressed.

1- It would be good if the authors explain some details about validity and reliability of the used questionnaires.

2- The manuscript does not clearly describe the sample method used (e.g., random) in the study to recruit participants.

Validity of the findings

1- The major issue with the study is the low number of participants and high withdraw rate which make it difficult to gain applicable clinical conclusion about the use of the app, where effectiveness needs to be evaluated for longer time.


2- Table 1 shows that most participants in the intervention group were above 50 years old. I think there is a significant difference between people who less than age of 50 and those who are over 50 in term of their experience, views and engagement of using such those apps. I would like a word about this in the Discussion section and whether this influence the study result or not.

3- There is no recommendation for future studies in this area. I would be interested to read your thoughts on future research in this area.

Additional comments

This study is quasi experimental study aimed to evaluate the effectiveness of an app on medication adherence to hypertension treatments and how users experience influence the outcomes. It is well conducted, but there are some issues that the authors need to be addressed.

·

Basic reporting

no comment

Experimental design

no comment

Validity of the findings

2 issues in the Conclusion:
1. Lines 246, 247 - "Regular monitoring by professionals encouraged users to pay more attention to their self-monitoring. " Even though the mHealth app contains an interaction function that enables patients directly communicating with healthcare providers, the authors did not provide any related results supporting this assumption (causation or correlation). I don't think the authors can draw this conclusion based on this study design.
2. Lines 249,250 - "This study also revealed that considering interaction aspects and design implications is imperative to achieve better health outcomes. " As I stated above.

Additional comments

This is a non-randomized open-label trial investigates the effect of the mHealth app on patients with hypertension. Previous studies demonstrated positive effects of mHealth over hypertension management (Medline: 32311957, Medline: 29710289, Medline: 30431384). Here I address several issues that I concern about:
1. The app was designed to record blood pressure, weight, waist circumference, height, sleep, mood, and engagement in physical activities. Is it possible to include these data, especially BP, in this manuscript to strengthen the conclusion?
2. Method, Line 153 -" Loss to follow-up was due to participants’ loss of interest and inability to provide data for analysis." What does "inability to provide data for analysis" mean? It appears to be the most significant reason that participants dropped out.
3. Discussion, Lines 191,192 - "Akoko et al. (2017) found that factors related to the patient and the health service providers showed significant associations with adherence. " what factors?

Reviewer 3 ·

Basic reporting

Introduction section
1. The statement in lines 34-36 needs to be more specific. I would suggest the authors specify which hypertension diagnostic threshold that they referred to when they argued that “one-third of patients with hypertension have not been diagnosed”. For example, the American Heart Association updated its hypertension diagnostic threshold from 140/90 mmHg to 130/80 mmHg. However, the new guideline has not been adopted in many developing countries. The following article documented the situation in China.
Ni, Z., He, J., Wang, J.G., Cao, J.P., Yang, Q., Wu, B., & Shaw, R.J. (2019). “Chinese
Physicians’ Perspectives on the 2017 American College of Cardiology/American Heart
Association Hypertension Guidelines”. High Blood Pressure & Cardiovascular Prevention,
26(3), 247-57. https://doi.org/10.1007/s40292-019-00321-9

2. The word “feasibility” in line 39 is not clear to me. The authors may need to rephrase the sentence – “Adherence to treatment relies on several factors, such as feasibility” – to make it easier to understand.

3. In the manuscript, “adherence to hypertension treatment” is a key concept. I would suggest the authors give a clear definition of it.

4. In the Introduction section, the authors listed several factors related to poor adherence to hypertension treatment, but in fact poor adherence to hypertension treatment is a complex issue relevant to a lot of factors, including patients’ age, educational level, number of prescribed medications, social-economic status, number of comorbidities, lack of awareness and etc. To strengthen the manuscript, more evidence should be studied, and more references should be cited to give a deeper and broader explanation of the reasons of poor adherence to hypertension treatment. The following is an article that reported several factors related to poor adherence to cardioprotective medications (including antihypertensive medications), which may help the authors to better understand why adherence to hypertension treatment is a public health issue.
Cardioprotective medication adherence among patients with coronary heart disease in China: a systematic review https://heartasia.bmj.com/content/11/2/e011173

5. The English language of this manuscript needs to be improved, particularly the third and fourth paragraphs of the Introduction section. For example, in lines 57-59, the sentence – “Technology such as mHealth apps can help patients remember to check blood pressure regularly, take medications and adopt healthy habits, all leading to better treatment adherence” – could be changed into “MHealth apps can remind patients of healthy habits, such as checking blood pressures regularly and taking medications as prescribed.” The sentence in lines 63-64 – “User experience studies can act as the force to investigate and explicate the needs and purposes of users of an mHealth app in providing the patient an exceptional health-care experience” – is difficult for me to understand.

6. In line 75, the sentence – “We conducted a quasi-experimental study (non-randomized, controlled, open-label) involving data collection before and after 12 weeks of follow-up” – is not clear to me. I would suggest the authors change it into “We conducted a quasi-experimental study (non-randomized, controlled, open-label) and collected participants’ data at enrollment and 12 weeks after our mHealth intervention.”

7. The first and fourth paragraphs in the Introduction section are not coherent. Also, in the fourth paragraph, the authors mentioned two concepts, “user experience” and “longitudinal approach”, but they were not described in a logical way and were a little difficult to understand.


Methods section
1. In lines 126-127, the sentence “During recruitment, we helped participants who had difficulties with the app, giving feedback and more detailed instructions.” is difficult for me to understand.

2. The authors mentioned that “the time frame was based on the study of Neumann et al (2015).” I would suggest the authors briefly mention the rationale that Neumann chose 12 weeks as the time frame.

3. The eligibility criteria of participants are ambiguous. In line 92, the criterion #1 is “current and ongoing medical monitoring and follow-up regarding hypertension treatment.” Who is going to monitor and follow-up? Healthcare providers or participants themselves? In line 94, it says that the criterion #3 is “ability to measure blood pressure periodically.” There are several ways of measuring blood pressure, such as using an electronic blood pressure cuff and a mercury sphygmomanometer. The authors should specify which measurement they refer to.

4. In the Statistical analysis section, the authors mentioned that “We analyzed the UEQ’s responses using its own data analysis tool”. What does “its own data analysis tool” mean?


Results section
1. In line 178, the phrase “to improve the preventive usefulness of the collected data” might need to be rephrased to make it easier for readers to understand.

2. In the Table 1, “20 to 29 years” may need to be changed into “20 to 39 years”.


Discussion section
1. The authors argued that “The age group of this study may have positively influenced the study’s adherence rate, according to the findings of Daniel and Veiga (2013), who observed that increasing age was associated with a higher probability of adherence to the recommended treatment.” This statement needs more evidence to support. Using one study’s finding to support that the age has positively influenced the study’s adherence rate is not convincing. For example, there are studies documented that increased age could decrease patients’ medication adherence.

2. The second paragraph of the Discussion section may need to be rephrased to improve its clarity. For example, “….to improve the preventive usefulness of the collected data….” is difficult for readers to understand.

Conclusion section
1. The Conclusion section should be more concise. Some of the content, such as “As better adherence implies several underlying improvements related to the diverse treatment factors, this can be considered a cornerstone of the success of digital interventions” could be moved to Discussion section.

Experimental design

no comment

Validity of the findings

no comment

Additional comments

This manuscript reported a non-randomized clinical trial in which an app was used to help patients with hypertension to improve adherence to hypertension treatment. This manuscript is a good start. To improve the quality of the manuscript, the authors could consider using university-based editorial services to improve the English language of the manuscript.

---

## Round 0.2 · Minor Revisions

I agree with your responses and modifications into the text, except for:

3. In the manuscript, “adherence to hypertension treatment”is a key concept. I would suggest the authors give a clear definition of it.


I would like to see a background about the definition of adherence in the relevant place.

·

Basic reporting

no comment

Experimental design

no comment

Validity of the findings

no comment

Additional comments

no comment

---

## Round 0.3 · accepted · Accept

All the reviewers' concerns have been correctly addressed.

Reviewer 1 ·

Basic reporting

The authors reported the study clearly with sufficient information.

Experimental design

The authors addressed all the provided comments

Validity of the findings

The authors did their best to address all provided comments

Additional comments

The authors addressed all the provided comments. Thank you.